# Controlling Over-generalization and its Effect on Adversarial Examples Generation and Detection

**Mahdieh Abbasi**
Université Laval
mahdieh.abbasi.1@ulaval.ca

**Arezoo Rajabi**
Oregon State University
rajabia@oregonstate.edu

**Azadeh Sadat Mozafari**
Université Laval
azadeh-sadat.mozafari.1@ulaval.ca

**Rakesh B. Bobba**
Oregon State University
rakesh.bobba@oregonstate.edu

**Christian Gagné**
Université Laval
christian.gagne@gel.ulaval.ca

## ABSTRACT

Convolutional Neural Networks (CNNs) significantly improve the state-of-the-art for many applications, especially in computer vision. However, CNNs still suffer from a tendency to confidently classify out-distribution samples from unknown classes into pre-defined known classes. Further, they are also vulnerable to adversarial examples. We are relating these two issues through the tendency of CNNs to over-generalize for areas of the input space not covered well by the training set. We show that a CNN augmented with an extra output class can act as a simple yet effective end-to-end model for controlling over-generalization. As an appropriate training set for the extra class, we introduce two resources that are computationally efficient to obtain: a representative natural out-distribution set and interpolated in-distribution samples. To help select a representative natural out-distribution set among available ones, we propose a simple measurement to assess an out-distribution set's fitness. We also demonstrate that training such an augmented CNN with representative out-distribution natural datasets and some interpolated samples allows it to better handle a wide range of unseen out-distribution samples and black-box adversarial examples without training it on any adversaries. Finally, we show that generation of white-box adversarial attacks using our proposed augmented CNN can become harder, as the attack algorithms have to get around the rejection (dustbin) regions when generating actual adversaries.

## 1 INTRODUCTION

Convolutional Neural Networks (CNNs) have allowed for significant improvements over the state-of-the-art in the last few years for various applications, and in particular for computer vision. Notwithstanding these successes, challenging issues remain with these models. In the following work, we specifically look at two concerns. First, CNNs are vulnerable to different types of *adversarial examples* (Szegedy et al., 2014; Kurakin et al., 2016b; Moosavi Dezfooli et al., 2016; Carlini and Wagner, 2017b). These adversarial examples are created by deliberately modifying clean samples with imperceptible perturbations, with the aim of misleading CNNs into classifying them to a wrong class with high confidence. Second, CNNs are not able to handle instances coming from outside the task domain on which they are trained — the so-called *out-distribution samples* (Liang et al., 2017; Lakshminarayanan et al., 2017). In other words, although these examples are semantically and statistically different from the (in-distribution) samples relevant to a given task, the neural network trained on the task assigns such out-of-concept samples with high-confidence to the pre-defined in-distribution classes. Due to the susceptibility of CNNs to both adversaries and out-distribution samples, deploying them for real-world applications, in particular for security-sensitive ones, is a serious concern.

These two issues have been treated separately in the past, with two distinct family of approaches. For instance, on the one hand, to handle out-distribution samples, some researchers have proposed threshold-based post-processing approaches with the aim of firstly calibrating the predictive confidence scores provided by either a single pre-trained CNN (Liang et al., 2017; Hendrycks and Gimpel, 2016; Lee et al., 2017) or an ensemble of CNNs (Lakshminarayanan et al., 2017), and then detecting out-distribution samples according to an optimal threshold. However, it is difficult to define an optimal and stable threshold for rejecting a wide range of out-distribution samples without increasing the false negative rate (i.e., rejecting in-distribution samples). On the other hand, researchers regarded adversarial examples as a distinct issue from the out-distribution problem and attempted to either correctly classify all adversaries through adversarial training of CNNs (Tramèr et al., 2017; Goodfellow et al., 2015; Moosavi Dezfooli et al., 2016) or reject all of them by training a separate detector (Feinman et al., 2017; Metzen et al., 2017). The performance of these approaches at properly handling adversarial instances mostly depends on having access to a diverse set of training adversaries, which is not only computationally expensive but also handling some possible future adversaries, which have not been discovered yet, most likely is difficult.

It is known that deep neural networks (e.g. CNNs) are prone to over-generalization in the input space by partitioning it *entirely* into a set of pre-defined classes for a given in-distribution set (task), regardless of the fact that in-distribution samples may only be relevant to a small portion of the input space (Liang et al., 2017; Spigler, 2017; Bendale and Boult, 2016). In this paper, we highlight that the two aforementioned issues of CNNs can be alleviated simultaneously through control of over-generalization. To this end, we propose that an *augmented CNN*, a regular (naive) CNN with an extra class dubbed as *dustbin*, can be a simple yet effective solution, if it is trained on appropriate training samples for the dustbin class. Furthermore, we introduce here a computationally-efficient answer to the following key question: how to acquire such an appropriate set to effectively reduced the over-generalized regions induced by naive CNN. We note that our motivation for employing an augmented CNN is different from the threshold-based post-processing approaches that attempt to calibrate the predictive confidence scores of a pre-trained naive CNN without impacting its feature space. Our motivation in fact is to learn a more expressive feature space, where along with learning the sub-manifolds corresponding to in-distribution classes, a distinct extra sub-manifold for the dustbin class can be obtained such that the samples drawn from many over-generalized regions including a wide-range of out-distribution samples and various types of adversaries are mapped to this "dustbin" sub-manifold.

As a training source for the extra class (dustbin), one can consider using synthetically generated out-distribution samples (Lee et al., 2017; Jin et al., 2017) or adversarial examples (Grosse et al., 2017). However, using such generated samples is not only computationally expensive but also barely able to effectively reduce over-generalization compared to naive CNNs (see Sec. 3). Instead of such synthetic samples, there are plenty of cost-effective training sources available for the extra dustbin class, namely *natural out-distribution datasets*. By natural out-distribution sets we mean the sets containing some realistic (not synthetically generated) samples that are semantically and statistically different from those in the in-distribution set. A *representative* natural out-distribution set for a given in-distribution task should be able to adequately cover the over-generalized regions. To recognize such a representative natural set, we propose a simple measurement to assess its fitness for a given in-distribution set. In addition to the selected set, we generate some artificial out-distribution samples through a straightforward and computationally efficient procedure, namely by interpolating some pair of in-distribution samples. We believe a properly trained augmented CNN can be utilized as a *threshold-free baseline* for identifying concurrently a broad range of unseen out-distribution samples and different types of strong adversarial attacks.

The main contributions of the paper are summarized as:

- By *limiting the over-generalization regions* induced by naive CNNs, we are able to drastically reduce the risk of misclassifying both adversaries and samples from a broad range of (unseen) out-distribution sets. To this end, we demonstrate that an *augmented CNN* can act as a simple yet effective solution.
- We introduce a measurement to select a *representative natural out-distribution set* among those available for training effective augmented CNNs, instead of synthesizing some dustbin samples using hard-to-train generators.
- Based on extensive experiments on a range of different image classification tasks, we demonstrate that properly trained augmented CNNs can significantly reduce the misclassification rates for

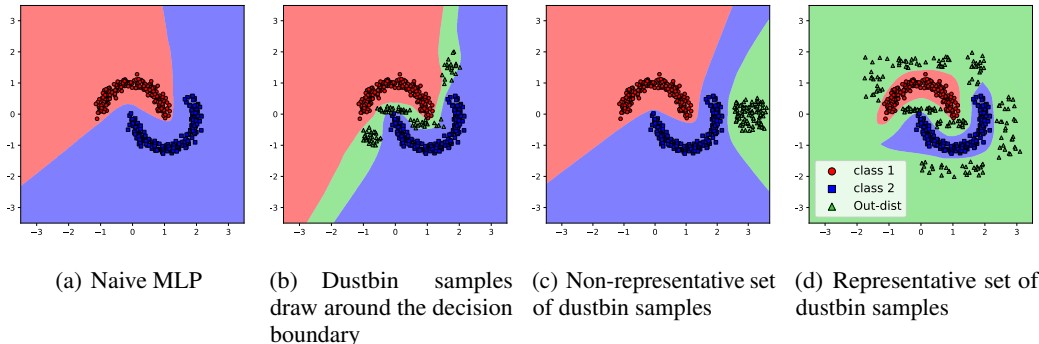

| (a) Naive MLP | (b) Dustbin samples draw around the decision boundary | (c) Non-representative set of dustbin samples | (d) Representative set of dustbin samples |

Figure 1: Illustration of the influence of different sets of dustbin training samples on the over-generalized regions. Two-moon classification dataset: (a) naive MLP trained only with in-distribution samples, (b-d) augmented MLPs trained with different out-distribution sets as dustbin. The MLP is made of three layers and ReLU activation functions.

both 1) *unseen out-distribution sets*, and 2) for various types of *strong black-box adversarial examples*, even though they are never trained on any specific types of adversaries.

- For the *generation of white-box adversaries* using our proposed augmented CNN, the adversarial attack algorithms frequently encounter dustbin regions rather than regions from other classes when distorting a clean samples, making the adversaries generation process *more difficult*.

## 2 PROPOSED METHOD

The key idea of this paper is to make use of a CNN augmented with a dustbin class, trained on a representative set of out-distribution samples, as a simple yet effective candidate solution to limit over-generalization. A visual illustration of this is given in Fig. 1, which provides a schematic explanation of the influence of training samples used to learn the dustbin class on the out-distribution area coverage.

This figure illustrates how the choice of training samples for the extra dustbin class plays a central role for achieving an effective augmented CNN. With a naive MLP (no dustbin class), a decision boundary is separating the whole input space into two classes (Fig. 1(a)), working on the complete input space even in regions that are deemed irrelevant for the task at hand. As for augmented MLPs, the second plot (Fig. 1(b)) shows results with dustbin training samples picked to be around the decision boundary, where many adversarial examples (Moosavi Dezfooli et al., 2016) are designed to be located. As it can be observed, such augmented MLP can only slightly reduce over-generalized regions. However, it might be able to classify some of adversaries as dustbin, and make generation of new adversaries harder since the adversarial attack algorithm should avoid the dustbin regions that are now located in between the two in-distribution classes. Thus, using solely such adversaries as training set of the dustbin class can not adequately cover the over-generalized regions. In another variation (Fig. 1(c)), the dustbin samples come from a out-distribution set, quite compact and located around the in-distribution samples from one specific class. Training an augmented MLP on this kind of out-distribution samples cannot reduce over-generalization effectively. Accordingly, we argue that out-distribution training samples that are distributed uniformly w.r.t in-distribution classes can be regarded as *a representative set* for the extra dustbin class (Fig. 1(d)). Indeed, an augmented MLP trained on a representative set is able to classify a wide-range of unseen out-distribution sets and some of adversaries as its extra class, being more effective at controlling over-generalization. It is worth to note that coupling a representative out-distribution set with the samples drawn around the decision boundaries can further strengthen the augmented neural network against adversarial examples.

There are many possible ways of acquiring some training samples for the extra class of augmented CNNs, ranging from artificially generated samples (Jin et al., 2017; Lee et al., 2017) to natural available out-distribution sets. Instead of making use of a generator, which is computationally expensive and hard to train, we propose the use of two cost-effective resources for acquiring dustbin

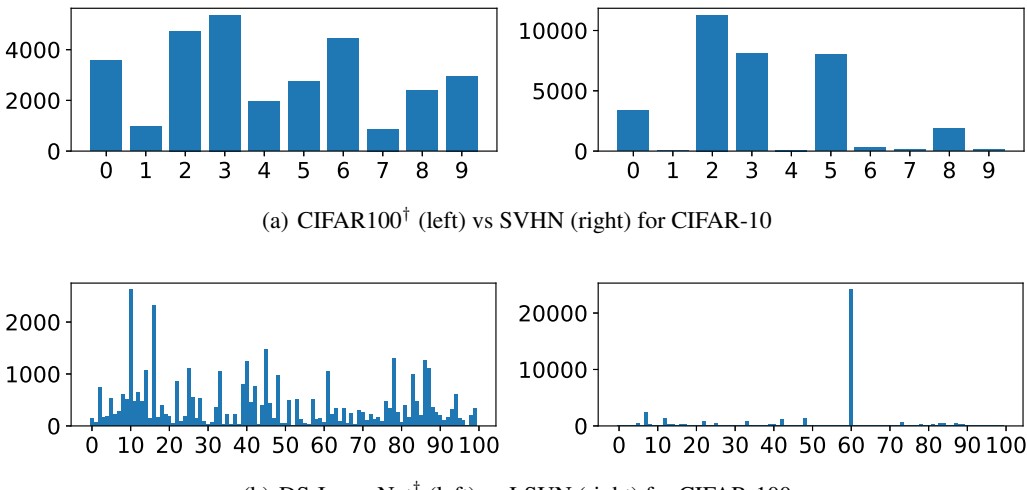

(a) CIFAR100$^\dagger$ (left) vs SVHN (right) for CIFAR-10

(b) DS-ImageNet$^\dagger$ (left) vs LSUN (right) for CIFAR-100

Figure 2: Misclassification distribution over original classes: (a) CIFAR-100$^\dagger$ vs SVHN provided by a naive VGG trained on CIFAR-10; (b) DS-ImageNet$^\dagger$ vs LSUN provided by a naive Resnet-164 trained on CIFAR-100.

training samples in order to train effective augmented CNNs: i) a selected representative natural out-distribution set and ii) interpolated samples.

## 2.1 NATURAL OUT-DISTRIBUTION INSTANCES

A possible rich and readily accessible source of dustbin examples for training augmented models lies in natural out-distribution datasets. These sets contain natural samples that are statistically and semantically different compared to the samples of a given task. For example, NotMNIST and Omniglot datasets can be regarded as natural out-distribution sets when trying to classify MNIST digits. However, it is not clear how to select a sufficiently representative set from the (possibly large) corpus of available datasets in order to properly train augmented CNNs. We shed light on the selection of a representative *natural out-distribution set* by introducing a simple visualization metric. Specifically, we deem a natural out-distribution set as representative for a given in-distribution task if it is misclassified uniformly over the in-distribution classes. That is, if roughly an equal number of out-distribution samples are classified confidently as belonging to each of the in-distribution classes by the naive neural network. Accordingly, to assess the appropriateness of out-distribution sets for a given task (or in-distribution set), we visualize the number of out-distribution samples that are misclassified to each of the in-distribution classes by using a histogram. In other words, a natural out-distribution set which has a more uniform misclassification distribution over the in-distribution classes appears better suited for training an effective augmented CNN.

In Fig. 2, the uniformity characteristics of SVHN vs CIFAR-100$^\dagger$ as out-distribution sets for CIFAR-10, and LSUN vs DS-ImageNet$^\dagger$ (i.e. Down Scaled ImageNet) for CIFAR-100 are shown[1]. According to Fig. 2(a), most of SVHN samples are misclassified into a limited number of CIFAR-10 classes (5 classes out of 10 classes), while CIFAR-100$^\dagger$ exhibits a relatively more uniform misclassifcation on CIFAR-10 classes. Therefore, compared with SVHN, we consider CIFAR-100 as a more representative natural out-distribution set for CIFAR-10. A full comparison of these two out-distribution sets according to their ability to control over-generalization can be found in Table 3 of the Appendix. Similar behaviour can also be observed for LSUN vs DS-ImageNet as two out-distribution resources for training an augmented Resnet164 on CIFAR-100 (as in-distribution). In this case DS-ImageNet$^\dagger$ has a more uniform distribution when compared with LSUN.

---

[1]Throughout the paper, $\dagger$ indicates a modified out-distribution set by discarding the classes that have exact or semantic overlaps with the classes of its corresponding in-distribution set. For example, the super-classes of vehicle from CIFAR-100 is removed due to their semantic overlap with automobile and truck classes of CIFAR-10. Refer to Appendix A for detail information.

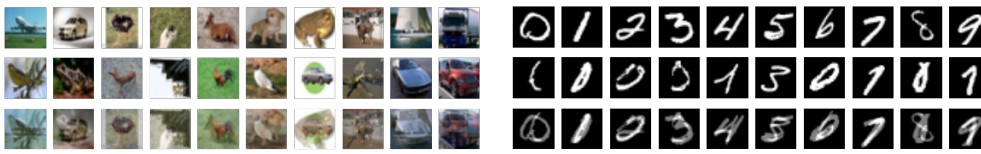

| (a) CIFAR10 interpolation | (b) MNIST interpolation |

Figure 3: Interpolated samples for CIFAR10 and MNIST. Third row for every dataset represents the interpolated samples that are composed of images from first (source) and second rows (target).

## 2.2 INTERPOLATED INSTANCES

Algorithms to generate adversarial examples tend to produce results near (on margin) decision boundaries separating two classes (Moosavi-Dezfooli et al., 2016). Adding a set of diverse types of adversaries to a representative natural out-distribution set may further improve the rate of adversary identification by the augmented CNN. But generating such a diverse set of adversarial examples for large-scale datasets is computationally expensive. Furthermore, using only adversaries as dustbin training samples (without including a representative natural out-distribution set) cannot lead to an effective reduction of over-generalization (see Fig. 1(b) and results in Sec. 3).

Instead of generating adversarial examples for training, we propose an inexpensive and straightforward procedure for acquiring some samples around the decision boundaries. To this end, we interpolate some pairs of correctly classified in-distribution samples from different classes. An interpolated sample created from two samples with different classes aims to cover such regions (margins around decision boundaries) between two classes in order to assign them to out-distribution (dustbin) regions. Formally speaking, consider a pair of input images from two different classes of a K-classification problem, i.e. $\mathbf{x}_i, \mathbf{x}_j \in \mathbb{R}^D$ $i, j \in \{1, ..., K\}$, where $\mathbf{x}_j$ is the nearest neighbor of $\mathbf{x}_i$ in the feature space of a CNN (its last convolution layer). An interpolated sample $\mathbf{x}' \in R^D$ is generated by making a linear combination of the given pair in the input space, $\mathbf{x}' = \alpha \mathbf{x}_i + (1 - \alpha) \mathbf{x}_j$. For all our experiments, we set $\alpha = 0.5$. Some interpolated samples can be seen in Fig. 3 for MNIST and CIFAR-10. The reasons for finding the nearest neighbors in the feature space are twofold: computationally less expensive yet more accurate (Bengio, 2009) when compared to doing so in high-dimensional input space.

## 2.3 FEATURE SPACE OF AUGMENTED CNNS

As an augmented CNN is trained in a end-to-end fashion, it allows learning of an extra sub-manifold corresponding to the added extra class (dustbin). Thus, if the augmented CNN is trained properly on a representative out-distribution set, it is able to map a large variety of out-distribution sets onto its extra sub-manifold, whether or not they have been seen during training. This should allow to learn a feature space that untangles the in-distribution set from the out-distribution samples. This is in contrast to the feature space of its naive counterpart, where the in-distribution and out-distribution samples are likely to be mixed or placed near each other.

Moreover, a proper trained augmented CNN is surprisingly able to map a large portion of black-box adversaries onto its extra manifold, even though it is never trained on any adversaries. Meanwhile some of the adversarial instances are mapped to their corresponding true class' sub-manifold. Therefore, this leads to a more engaging classifier for many practical situations (real-world applications) as some of adversaries are classified into dustbin (equivalent to the rejection option) while some of remaining ones are correctly classified as their true class (particularly non-transferable adversaries attacks, see Sec.3).

In Fig. 4, we exhibit the feature spaces achieved from a naive CNN and its augmented counterpart for CIFAR-10 as an in-distribution task. Note CIFAR-100 is used as the training set for the extra class of the augmented CNN. As it can be visualized in Fig. 4, the two out-distribution sets, including CIFAR-100 (green triangles) and Fast Gradient Sign (FGS) adversaries (Goodfellow et al., 2014) (shown with yellow triangles) are separated from CIFAR-10 samples in the feature space of the augmented CNN while they are mixed in the feature space of its naive counterpart.

| Naive CNN | Augmented CNN |

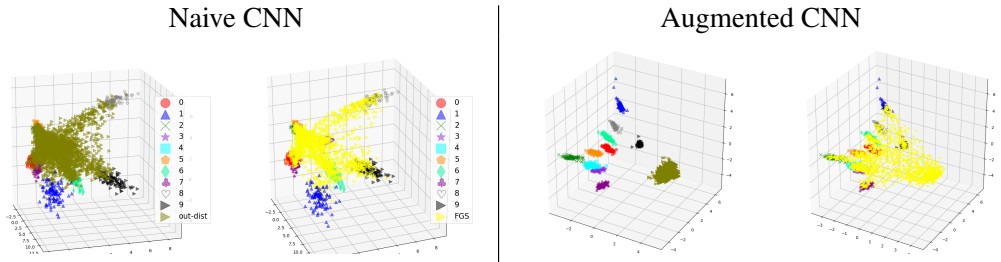

Figure 4: Visualization of data distribution in last convolution layer (i.e., feature space) of an augmented CNN trained on CIFAR-10 and CIFAR-100 as in-distribution and out-distribution sets, respectively. For visualization purposes, these feature spaces are reduced to 3D using PCA. For more results, refer to Fig 7 of the Appendix.

## 3 EVALUATION

We conduct several experiments on three benchmarks, namely MNIST, CIFAR-10, and CIFAR-100 datasets, using three neural network architectures LeNet (LeCun et al., 1998), VGG-16 (Simonyan and Zisserman, 2014), and ResNet164 (He et al., 2016). To assess robustness of the augmented versions of these CNNs , we consider five well-known strong attack algorithms: Fast Gradient Sign (FGS) (Goodfellow et al., 2014), Iterative FGS (I-FGS) (Madry et al., 2017), Targeted FGS (T-FGS) (Kurakin et al., 2016a), DeepFool (Moosavi Dezfooli et al., 2016), and C&W (Carlini and Wagner, 2017b) (see Appendix A.6 to learn about their hyper-parameter configurations). Note that we evaluate performance using three metrics: 1) accuracy (Acc.), which captures the rate or percentage of samples classified correctly as their true associated label; 2) rejection rate (Rej.), to measure the rate of samples correctly classified as dustbin (equivalent to rejection option); and 3) error rate (Err.), which captures the rate of samples that are neither correctly classified nor rejected.

### 3.1 BLACK-BOX ADVERSARIAL EXAMPLES

It is widely known that many of adversarial examples generated from a learning model (e.g., CNN) can be transferred to attack other victim models (Papernot et al., 2017; Szegedy et al., 2014; Carlini and Wagner, 2017a) — such attacks are called *transferable black-box attacks*. To evaluate robustness of the augmented CNNs on the aforementioned types of attacks generated in black-box setting, we generate adversarial samples corresponding to correctly classified clean test samples using a naive CNN, trained with different initial weights compared to the one under evaluation. Moreover, in order to demonstrate the influence of using different out-distribution sets for training the extra class on identifying adversaries, we employ four different sources for acquiring dustbin training samples: 1) adversarial samples generated by I-FGS; 2) only interpolated in-distribution data; 3) only a representative natural out-distribution set (selected according our proposed metric); and 4) both interpolated samples along with a representative natural out-distribution set (selected according our proposed metric).

To evaluate the generalization performance of the augmented CNNs on the in-distribution tasks, the in-distribution test accuracy rates are presented in Table 1. Compared to the naive CNNs, we observe a slight drop in test accuracy rates of their augmented counterparts (except for that trained on I-FGS adversaries) while, interestingly, having also the error rates (i.e., the number of wrong decisions) reduced, leading to less error in decision making. This property can be highly beneficial for some security-sensitive applications, where making less error in some critical situations is vital.

For the augmented CNNs, rejection rate (i.e., assignments to dustbin) is reported in addition to accuracy (i.e., correct classifications) and error rates (i.e., misclassifications)[2]. Comparing the augmented CNNs in Table 1 across different classification tasks, we can find that the augmented

---

[2]From Table 1 we can obtain out-distribution detection performance, with true positive rate as (in-distribution accuracy + error rate), false positive rate as (1 - out-distribution rejection rate), and false negative as in-distribution rejection rate. As the proposed approach is threshold-free, ROC curves and related measurements (e.g., AUC-ROC) are irrelevant.

| | | | Naive Models | Augmented Models | | | |
|---|---|---|---|---|---|---|---|
| | | | | Adversarial (I-FGS) | Out-distribution | Interpolation | Out-dist. + Interp. |
| **MNIST / NotMNIST (LeNet)** | In-dist. (MNIST) test | Acc. | 99.50 | 99.54* | 99.47 | 99.50 | 99.48 |
| | | Rej. | — | 0.00 | 0.02 | 0.02 | 0.08 |
| | | Err. | 0.50 | 0.46 | 0.51 | 0.48 | 0.44 |
| | Out-dist. (NotMNIST) test | Rej. | — | 80.75 | 99.96 | 47.97 | **99.98** |
| | FGS | Acc. | 35.14* | 0.00 | 19.15 | 10.47 | 0.34 |
| | | Rej. | — | 100.00 | 65.19 | 83.31 | 99.59 |
| | | Err. | 65.86 | **0.00** | 15.66 | 6.22 | 0.07 |
| | I-FGS | Acc. | 25.90 | 0.00 | 39.20* | 3.10 | 0.01 |
| | | Rej. | — | 100.00 | 23.20 | 95.69 | 99.90 |
| | | Err. | 74.10 | **0.00** | 37.60 | 1.21 | 0.09 |
| | T-FGS | Acc. | 19.99* | 0.00 | 1.17 | 1.05 | 0.00 |
| | | Rej. | — | 100.00 | 95.92 | 98.58 | 100.00 |
| | | Err. | 80.01 | **0.00** | 0.37 | **0.00** | **0.00** |
| | DeepFool | Acc. | 1.89 | 6.21 | 11.45* | 9.87 | 5.36 |
| | | Rej. | — | 35.66 | 4.72 | 78.06 | 89.84 |
| | | Err. | 98.11 | 58.13 | 83.83 | 12.07 | **4.80** |
| | C&W ($L_2$) | Acc. | 22.49 | 18.00 | 27.50* | 15.50 | 7.50 |
| | | Rej. | — | 28.00 | 5.99 | 55.00 | 77.49 |
| | | Err. | 77.51 | 54.00 | 66.51 | 29.50 | **15.01** |
| **CIFAR-10 / CIFAR-100† (VGG)** | In-dist. (CIFAR-10) test | Acc. | 90.53 | 91.66* | 88.58 | 90.38 | 86.65 |
| | | Rej. | — | 0.10 | 5.69 | 1.55 | 8.74 |
| | | Err. | 9.47 | 8.24 | 5.73 | 8.07 | **4.61** |
| | Out-dist. (CIFAR-100†) test | Rej. | — | 0.82 | 95.36 | 5.00 | **96.21** |
| | FGS | Acc. | 36.16 | 0.00 | 31.90 | 38.98* | 29.50 |
| | | Rej. | — | 100.00 | 36.81 | 6.93 | 45.11 |
| | | Err. | 63.84 | **0.00** | 31.29 | 54.09 | 25.39 |
| | I-FGS | Acc. | 51.19 | 0.00 | 54.27 | 59.22* | 50.28 |
| | | Rej. | — | 100.00 | 12.94 | 5.06 | 24.76 |
| | | Err. | 48.81 | **0.00** | 32.79 | 35.72 | 24.96 |
| | T-FGS | Acc. | 36.24* | 0.00 | 27.17 | 33.51 | 24.35 |
| | | Rej. | — | 100.00 | 41.08 | 7.32 | 51.33 |
| | | Err. | 63.76 | **0.00** | 31.75 | 59.17 | 24.32 |
| | DeepFool | Acc. | 56.82* | 46.52 | 44.22 | 54.48 | 42.81 |
| | | Rej. | — | 14.12 | 33.20 | 5.16 | 40.26 |
| | | Err. | 43.18 | 39.36 | 22.58 | 40.36 | **16.93** |
| | C&W ($L_2$) | Acc. | 42.50 | 44.50 | 46.50 | 48.50* | 39.00 |
| | | Rej. | — | 1.50 | 18.50 | 8.00 | 39.50 |
| | | Err. | 57.50 | 54.00 | 35.00 | 43.50 | **21.50** |
| **CIFAR-100 / DS-ImageNet† (ResNet164)** | In-dist. (CIFAR-100) test | Acc. | 75.52* | 75.33 | 74.75 | 73.68 | 73.37 |
| | | Rej. | — | 0.08 | 0.93 | 4.85 | 5.02 |
| | | Err. | 24.48 | 24.59 | 24.32 | 22.44 | **21.61** |
| | Out-dist. (DS-ImageNet†) test | Rej. | — | 12.11 | 97.24 | 3.28 | **97.33** |
| | FGS | Acc. | 67.67 | 6.05 | 50.15 | 67.77* | 50.03 |
| | | Rej. | — | 92.17 | 34.22 | 10.46 | 36.87 |
| | | Err. | 32.33 | **1.78** | 15.63 | 21.77 | 13.10 |
| | I-FGS | Acc. | 22.20 | 0.00 | 16.90 | 26.55* | 16.80 |
| | | Rej. | — | 100.00 | 32.65 | 18.45 | 45.75 |
| | | Err. | 77.80 | **0.00** | 50.45 | 55.00 | 37.45 |
| | T-FGS | Acc. | 59.93* | 2.55 | 37.87 | 57.48 | 37.07 |
| | | Rej. | — | 95.65 | 44.12 | 14.41 | 46.87 |
| | | Err. | 40.07 | **1.80** | 11.76 | 28.11 | 16.06 |
| | DeepFool | Acc. | 77.20* | 73.77 | 67.12 | 73.02 | 66.27 |
| | | Rej. | — | 0.95 | 10.56 | 5.85 | 15.32 |
| | | Err. | 22.80 | 25.28 | 22.32 | 21.13 | **18.41** |
| | C&W ($L_2$) | Acc. | 74.50* | 68.00 | 66.00 | 68.00 | 60.50 |
| | | Rej. | — | 4.00 | 16.50 | 11.00 | 25.50 |
| | | Err. | 25.50 | 28.00 | 17.50 | 22.00 | **14.00** |

Table 1: Results for black-box adversaries attacks on three classification tasks. Values with $^*$ denotes best accuracy while **boldface** denotes lowest misclassification rate for each given dataset and attack method.

CNNs trained on a set of I-FGS adversaries can reject (classifying as dustbin) almost all test variants of FGS adversaries (i.e., FGS, I-FGS and T-FGS), however they fail to reject non-FGS variants of adversaries (e.g., C&W and DeepFool), as well as the natural out-distribution sets (see Table 4 of Appendix A). *Accordingly, we emphasize that using the samples drawn from the vicinity of decision boundaries such as I-FGS adversaries as a single training source for the extra dustbin class of augmented CNN can not effectively control over-generalization.* Contrary to I-FGS augmented CNN, augmented CNNs trained on a representative natural out-distribution set (selected according to our proposed metric) along with some interpolated samples consistently outperform their naive counterparts and the other augmented CNNs by achieving a drastic drop in error (misclassification) rates on all variants of adversaries, even though these augmented CNNs are not trained on any specific type of adversaries. This illustrates that if an augmented CNN is trained on a representative out-distribution set along with some interpolated samples, it can efficiently reduce over-generalization, resulting in generally well-performing model in the case of adversaries and various natural out-

distribution samples. Due to space limitation, we place some results on the augmented CNNs trained with non-representative natural out-distribution sets in the Appendix A in Table 3 for illustrating the deficiency of such sets in controlling over-generalization.

To visualize and compare the classification regions in input space of our augmented CNN and its naive counterpart, we plot several church-windows (cross-sections) (Warde-Farley and Goodfellow, 2016) in Fig. 5. The x-axis of each window is the adversary direction achieved by FGS or DeepFool using the naive network. For each adversary direction, we plot four windows by taking four random directions that are perpendicular to the given adversary direction (x-axis). As it can be observed, the fooling classification regions (spanned by the adversary direction and one of its orthogonal random directions) of the naive CNNs are occupied by dustbin regions (indicated by orange) in their augmented counterparts.

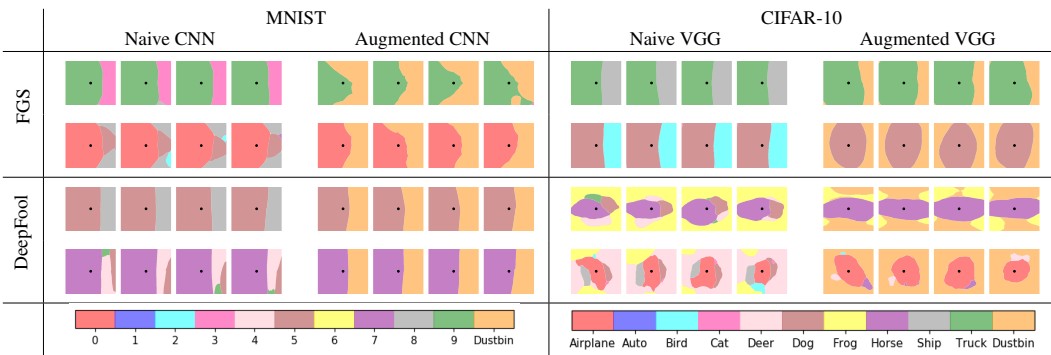

Figure 5: Church window plots for various data instances. Black dot corresponds to the clean sample position.

## 3.2 WHITE-BOX ADVERSARIAL EXAMPLES

White-box adversarial examples are generated by using directly the model on which they are applied. We further evaluate the robustness of our augmented CNNs on different types of white-box attacks, using the same parameter configurations as with the black-box experiments. For this purpose, we compute the percentage of visiting fooling classes (i.e., the classes different from dustbin and the true class associated to the clean samples) and the dustbin class when moving in the direction given by an attack method for a set of clean samples. Note that for generating some authentic white-box adversaries by the augmented CNNs, the attack algorithm should avoid dustbin regions to preclude generation of useless adversaries (those already recognizable as dustbin by the augmented CNN). In addition, the percentage of visiting the dustbin class when moving in a "legitimate" direction is also reported. By this we mean moving from a given sample $\mathbf{x}$ to its nearest neighbor $\mathbf{x}'$ from the *same class* in the direction of their convex combination $(1 - \epsilon)\mathbf{x} + \epsilon\mathbf{x}'$. Results for legitimate directions are computed with varying $\epsilon \in [0.1, 0.5]$.

To generate white-box adversaries using both a naive CNN and its augmented counterpart, MNIST and CIFAR-10 test sets are utilized. As seen in Fig. 6, adversaries generated for the augmented CNNs (trained on a representative natural out-distribution set) encounter more often the dustbin class than a fooling class, indicating that generation of white-box adversaries using the augmented CNNs becomes harder. An adversarial algorithm needs to skip over some regions assigned to dustbin class, leading to a possible increase in the number of steps or the amount of distortions required for generating adversaries. Moreover, by moving in legitimate directions, the augmented CNNs appear to remain largely in the current true classes.

## 3.3 OUT-DISTRIBUTION SAMPLES

The behavior of augmented CNNs is evaluated on several out-distribution sets across different in-distribution tasks. For each in-distribution task, we consider several natural out-distribution datasets, both seen and unseen during the training of the augmented CNN. For comparison purposes, the rejection rates of two recent threshold-based approaches, including ODIN (Liang et al., 2017), and

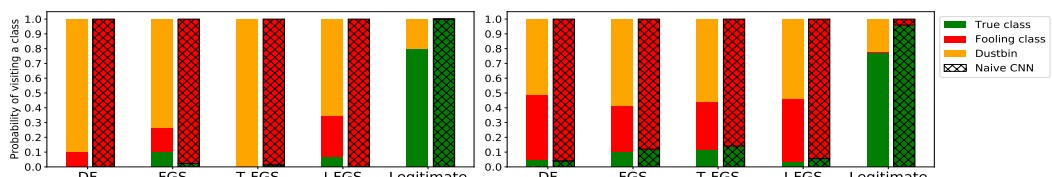

Figure 6: Robustness of naive CNNs (LeNet for MNIST (left) and VGG-16 for CIFAR-10 (right)) and their augmented counterparts under different adversarial attack algorithms. Robustness to white-box attacks is measured by the percentage of visiting fooling classes and dustbin class by moving in adversarial directions.

| In-distribution train | Out-distribution test | Naive model Error (%) | Augmented model Error (%) | Rejection (%) | ODIN Rejection (%) | Calibrated CNN Rejection (%) |
|---|---|---|---|---|---|---|
| MNIST | NotMNIST (seen) | 93.15 | 0.01 | 99.98 | 61.34 | **100** |
| | Omniglot* (unseen) | 95.19 | 0.00 | **100** | 96.70 | **100** |
| | CIFAR-10(gc) (unseen) | 64.26 | 0.00 | **100** | 99.51 | 99.8 |
| CIFAR-10 | CIFAR-100† (seen) | 97.05 | 3.71 | **96.21** | 48.1 | 29.64 |
| | DS-ImageNet† (unseen) | 96.62 | 12.20 | **87.49** | 54.27 | 79.63 |
| | SVHN (unseen) | 95.56 | 7.61 | **92.29** | 60.68 | 57.42 |
| | LSUN* (unseen) | 96.12 | 14.31 | **84.80** | 69.26 | 71.5 |
| CIFAR-100 | DS-ImageNet† (seen) | 79.34 | 1.52 | **98.35** | 32.06 | 15.77 |
| | SVHN (unseen) | 81.19 | 67.75 | 16.25 | **51.65** | 7.5 |
| | LSUN* (unseen) | 96.12 | 0.01 | **99.99** | 41.16 | 13.7 |

Table 2: Comparison of augmented CNNs and threshold-based approaches on a range of natural out-distribution sets. The size of input images of the datasets indicated by * are scaled to be consistent with their corresponding in-distribution set. CIFAR-10 (gc) means gray-scaled and cropped version of CIFAR-10.

Calibrated CNN (Lee et al., 2017) are considered[3]. These approaches attempt to identify and reject out-distribution samples according to a specific threshold on the calibrated predictive confidence scores. For a fair comparison, the rejection rates (i.e. True Negative Rate) of these approaches are reported at the same True Positive Rates (TPR) as ours, where TPRs are considered $99\%$, $91\%$, and $95\%$ for MNIST, CIFAR-10, and CIFAR-100, respectively.[4]

Moreover, for all ODIN experiments, we consider $T = 1000$ and $\epsilon \in \{0, 5 \times 10^{-6}, 5 \times 10^{-5}, 5 \times 10^{-4}, 5 \times 10^{-3}, 1 \times 10^{-3}\}$ is tuned for each pair of in-distribution and out-distribution validation sets such that the highest possible TNR at the specified TPR can be achieved. For "calibrated CNN" approach, its hyper-parameter ($\beta > 0$), which controls the effect of having the calibrated (uniform) predictions on the synthesized out-distribution samples, is tuned such that training of the calibrated CNN can be converged on the given in-distribution training set. We observe while the larger beta for MNIST and CIFAR-10 lead to better calibrated CNN on out-distribution samples, such a large beta for CIFAR-100 does not allow its training to converge. Considering this trade-off between calibration and convergence, in our experiments, the values of $\beta$ are regarded 1 and 0.01 for CIFAR-10/MNIST and CIFAR100, respectively.

Table 2 compares the rejection rates (i.e TNR) of ours with ODIN and "calibrated CNN" as well as the error rates of naive and the augmented CNNs (trained on a representative natural out-distribution set and interpolated samples), where the error rate measures the number of the out-distribution samples classified with confidence higher than $50\%$ as one of the in-distribution classes. These error rates by naive CNN aim to show the fact that a significant portion of out-distribution samples are confidently (confidence$> 50\%$) misclassified by naive CNN. As it can be seen in Table 2, the augmented CNNs, which is trained on one single but representative natural out-distribution set (as well as interpolated samples), almost outperforms "calibrated CNN", which is trained on a set of synthetic out-distribution samples, and ODIN, which its hyper-parameter $\epsilon$ is tuned for each pair of in-distribution and out-distribution validation set. It can demonstrate how controlling effectively over-generalization can lead to developing more robust CNNs in the presence of novel unseen out-distribution sets.

---

[3]The code made available on Github by the authors of those papers is used for our evaluation.

[4]Note as our approach is independent of a threshold, we have only one fixed TPR for a given in-distribution set, which is computed from (Acc in-dist. + err. in-dist)

## 4 Conclusion

In this paper we bridge two issues of CNNs that were previously thought of as unrelated: susceptibility of naive CNNs to various types of adversarial examples and incorrect high confidence prediction for out-distribution samples. We argue these two issues are connected through over-generalization. We propose augmented CNNs as a simple yet effective solution for controlling over-generalization, when they are trained on an appropriate set of dustbin samples. Through empirical evidence, we define an indicator for selecting an "appropriate" natural out-distribution set as training samples for dustbin class from among those available and show such selection plays a vital role for training effective augmented CNNs. Through extensive experiments on several augmented CNNs in different settings, we demonstrate that reducing over-generalization can significantly reduce the misclassification error rates of CNNs on adversaries and out-distribution samples, simultaneously, while their accuracy rates on in-distribution samples are maintained. Indeed, reducing over-generalization by such an end-to-end learning model (e.g., augmented CNNs) leads to learning more expressive feature space where these two categories of hostile samples (i.e., adversaries and out-distribution samples) are disentangled from in-distribution samples.

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

# A   APPENDIX

## A.1   DETAILS ON DATASETS AND EXPERIMENTS

**MNIST with NotMNIST**   MNIST consists of gray scale images of hand-written digits (0-9) and is made of 60k and 10k samples for training and testing, respectively. NotMNIST dataset[5], which involves 18,724 letters (A-J) printed with different font styles, is used as a source of out-distribution samples for MNIST. Images of both MNIST and NotMNIST datasets have the same size ($28 \times 28$ pixels), with all pixels scaled in $[0, 1]$. LeNet, the CNN model we used comprised three convolution layers of 32, 32, and 64 filters ($5 \times 5$), respectively, and one Fully Connected (FC) layer with softmax activation function[6]. In addition, dropout with $p = 0.5$ is used on the FC layer for regularization. The augmented version of LeNet is trained with the 50k samples of MNIST, $10K$ randomly selected samples from NotMNIST for out-distribution samples and $15K$ interpolated samples (see Section2) generated from MNIST training samples. The remaining samples from NotMNIST ($\approx$8K) are used together with MNIST test samples to evaluate the augmented CNN.

**CIFAR-10 with CIFAR-100**[†]   CIFAR-10 and CIFAR-100 represents low-resolution RGB images ($32 \times 32$) of objects. CIFAR-10 contains 50k training and 10k testing instances over 10 classes. CIFAR-100 has the same characteristics except it is organized into 100 classes. For the experiments with CIFAR-10, out-distribution samples are taken from CIFAR-100[†]. To avoid the semantic overlaps between the labels of CIFAR-10 and CIFAR-100, super-classes of CIFAR-100 conceptually similar to those of CIFAR-10 are ignored (i.e., vehicle 1, vehicle 2, medium-sized mammals, small mammals, and large carnivores excluded from CIFAR-100). Pixels are scaled in $[0, 1]$, and then normalized by subtracting the mean of the image of the CIFAR-10 training set. VGG-16 (Simonyan and Zisserman, 2014) is used as CNN architecture for CIFAR-10, which has has 13 convolution layers of $3 \times 3$ filters and three FC layers. To train the augmented VGG-16, $15k$ samples are selected from CIFAR-100[†] along with $15k$ interpolated samples from CIFAR-10 training set (both labeled as dustbin) and are appended to the CIFAR-10 training set.

**CIFAR-100 with DS-ImageNet**[†]   Similar to CIFAR-10, training and test sets of CIFAR-100 contain 50K and 10K RGB images ($32 \times 32$ pixels each). As out-distribution samples for CIFAR-100, we utilized down-scaled version of ImageNet dataset (called DS-ImageNet) (Chrabaszcz et al., 2017) (images are scaled to $32 \times 32$). To choose proper out-distribution samples for CIFAR-100, we utilized the samples from 62 classes of DS-ImageNet that have less conceptual overlap with CIFAR-100 labels. For creating the training set for out-distribution dustbin class, samples from those 62 classes are taken from training set of DS-ImageNet. Therefore, this training set has 79,856 samples in total, but we randomly selected 15K of them along with 15K interpolated samples to train our augmented CNNs. We utilized validation set of DS-ImageNet containing 50K images as the test out-distribution task. We use ResNet-164 (He et al., 2016) to train an augmented CNN on CIFAR-100 (as in-distribution task).

## A.2   SELECTING NON REPRESENTATIVE OUT-DISTRIBUTION SET

For a given in-distribution dataset, there are many possible candidate out-distribution datasets for training an augmented CNN. We argued in section 2 that an non-representative natural out-distribution set, can not effectively handle over-generalization. According to our measurement, a non-representative natural out-distribution set is the the one that are only miscalssified as a limited-number of in-distribution classes by a nive CNN. Recall that, according to Fig. 2(a), most of SVHN samples are misclassified into a limited number of CIFAR-10 classes (5 classes out of 10 classes) by the naive CNN, while CIFAR-100[†] dataset exhibits a relatively more uniform misclassifcation on CIFAR-10 classes. Therefore, compared to SVHN, we consider CIFAR-100[†] as a more representative natural out-distribution set for CIFAR-10. Similarly, for CIFAR-100, DS-ImageNet[†] dataset is more uniformly misclassified when compared with LSUN and is thus considered a more appropriate out-distribution dataset.

In Table 3, we compare two types of out-distribution sets, representative vs non-representative across two classification tasks, and showing choosing a representative out-distribution set is a key factor for effectively reducing over-generalization such that a wide-range of other unseen out-distribution samples and adversarial examples can be confidently classified as dustbin (equivalent to rejection).

In comparison to the augmented VGG used SVHN, the augmented VGG-16 trained on CIFAR-100[†] as out-distribution training samples performs significantly better at rejecting both adversaries and unseen out-distribution samples (Table 3). Similarly, when comparing two augmented Resnets (for CIFAR-100 as in-distribution), the one trained with LSUN as the source of out-distribution samples is less effective in reducing over-generalization when comparison to the other Resnet trained with DS-ImageNet[†] as the source of out-distribution samples.

---

[5]Available at `http://yaroslavvb.blogspot.ca/2011/09/notmnist-dataset.html`.
[6]Details on the configuration are given in `https://github.com/dnouri/cuda-convnet/blob/master/example-layers/layers-18pct.cfg`.

| | Out-dist. Training -> | | CIFAR-10 (VGG-16) | | CIFAR-100 (Resnet-164) | |
|---|---|---|---|---|---|---|
| | | | SVHN | CIFAR-100† | LSUN | DS-ImageNet† |
| In-dist. Test | | Acc. | **91.71** | 88.58 | 74.68 | **74.75** |
| | | Rej. | 0.07 | 5.69 | 0.03 | 0.93 |
| | | Err. | 8.22 | **5.73** | 25.29 | **24.32** |
| Out-dist. Test | SVHN | Rej. | 99.94 | 93.44 | 0 | 23.61 |
| | | Err. | **0.06** | 6.36 | 81.46 | **63.65** |
| | LSUN | Rej. | 0 | 85.72 | 100 | 100 |
| | | Err. | 92.71 | **13.70** | 0 | 0 |
| | CIFAR-100† | Rej. | 3.82 | 95.30 | – | – |
| | | Err. | 92.27 | **4.52** | – | – |
| | DS-ImageNet† | Rej. | 0.55 | 65.00 | 28.76 | 97.37 |
| | | Err. | 95.36 | **33.93** | 55.33 | **2.46** |
| Adversarial Examples | FGS | Acc. | 37.65 | 31.90 | 70.28 | 50.15 |
| | | Rej. | 0.0 | 36.81 | 2.76 | 33.85 |
| | | Err. | 62.35 | **31.29** | 26.96 | **16** |
| | I-FGS | Acc. | 52.50 | 54.27 | 29.75 | 16.90 |
| | | Rej. | 0.0 | 12.94 | 1.85 | 32.65 |
| | | Err. | 47.50 | **32.79** | 68.4 | **50.45** |
| | T-FGS | Acc. | 31.28 | 27.17 | 60.48 | 37.87 |
| | | Rej. | 0.05 | 41.08 | 4.30 | 44.12 |
| | | Err. | 68.67 | **31.75** | 35.22 | **18.01** |
| | DeepFool | Acc. | 53.88 | 44.22 | 75.73 | 67.12 |
| | | Rej. | 0.05 | 33.20 | 1.05 | 10.56 |
| | | Err. | 46.07 | **22.58** | 23.22 | **22.32** |
| | C&W ($L_2$) | Acc. | 47.00 | 46.50 | 75.50 | 66.00 |
| | | Rej. | 7.00 | 18.50 | 0.5 | 16.50 |
| | | Err. | 46 | **35** | 24 | **17.5** |

Table 3: Comparing the performance of augmented CNNs trained on a representative v.s. non-representative out-distribution sets.

## A.3 ADVERSARIAL TRAINING OF AUGMENTED CNN

Here, we provide more results on the performance of augmented CNNs trained on the adversarial (I-FGS) samples labeled as dustbin to reject out-distribution samples. Although such augmented CNNs can reject perfectly the variants of FGS adversaries, they are not able to significantly reduce over-generalization as their error rates on a wide-range of out-distribution samples are considerably high (*e.g.,* 94.77% error rate for CIFAR-100 images on CIFAR-10 classifier).

| In-distribution train | Out-distribution test | Augmented CNN (I-FGS) | |
|---|---|---|---|
| | | Error (%) | Rejection (%) |
| MNIST | NotMNIST (unseen) | 18.31 | 80.75 |
| | Omniglot (unseen) | 0 | 100.0 |
| | CIFAR-10(gc) (unseen) | 0.09 | 99.89 |
| CIFAR-10 | CIFAR-100† (unseen) | 94.77 | 0.82 |
| | DS-ImageNet† (unseen) | 82.93 | 12.08 |
| | SVHN (unseen) | 94.56 | 0.04 |
| | LSUN (unseen) | 75.46 | 15.90 |
| CIFAR-100 | DS-ImageNet† (unseen) | 67.30 | 12.11 |
| | SVHN (unseen) | 82.51 | 0.09 |
| | LSUN (unseen) | 47.05 | 26.28 |

Table 4: The performance of adversarial training of augmented CNN on a wide-range of unseen adversarial examples.

## A.4 ODIN PERFORMANCE ON BLACK-BOX ADVERSARIES

Although ODIN's main concentration is on identifying only out-distribution samples, we compare it with ours on black-box adversarial examples in Table 5. For obtaining ODIN results, the naive pre-trained models that used in Table 1 (having the same architecture as our augmented CNNs except the number of outputs) are considered and the optimal value for the hyper-parameters of ODIN, i.e. $\epsilon \in \{0, 5 \times 10^{-6}, 5 \times 10^{-5}, 5 \times 10^{-4}, 5 \times 10^{-3}, 1 \times 10^{-3}\}$ and $T = 1000$ are utilized. The optimal value for $\epsilon$ is chosen separately for each set of black-box adversaries. For a fair comparison with ours, the ODIN results are reported at True Positive Rate

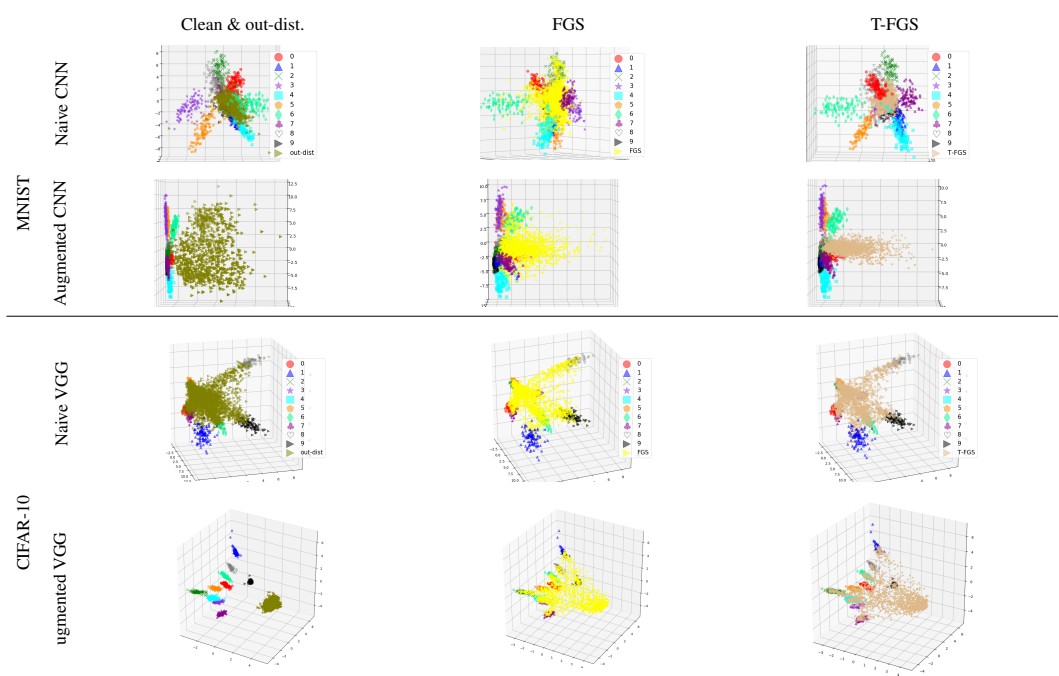

Figure 7: Visualization of some randomly selected test samples and their corresponding adversaries (FGS and T-FGS) in the feature spaces (the penultimate layer) learned by a naive CNN and an augmented CNN. To produce and manipulate the 3D plots refer to `https://github.com/mahdaneh/Out-distribution-learning_FSvisulization`

(TPR) 91% and 95% for CIFAR-10 and CIFAR-100, respectively. As it can be seen from Table 5, the error rates of our augmented CNN on most of the various types of black-box adversaries are lower than ODIN due to its higher adversaries rejection (classifying as dustbin) rates. For example, our method on I-FGS adversaries of CIFAR-100, which are highly transferable (it can be perceived from naive ResNet-164's low accuracy on I-FGS (i.e. 22.20% from Table 1)), has error rate 37.45% as it rejects 45.75% of this I-FGS adversaries, while ODIN rejects only 5.75% of adversaries, resulting to error rate 73.7%.

| Dataset | Method | FGS | | | I-FGS | | | T-FGS | | | DeepFool | | | C&W ($L_2$) | | |
|---------|--------|------|------|------|-------|------|------|-------|------|------|----------|------|------|-------------|------|------|
| | | Acc. | Rej. | Err. | Acc. | Rej. | Err. | Acc. | Rej. | Err. | Acc. | Rej. | Err. | Acc. | Rej. | Err. |
| CIFAR-10 | Ours | 29.50 | 45.11 | **25.39** | 50.28 | 24.76 | 24.96 | 24.35 | 51.33 | **24.33** | 42.81 | 40.26 | **16.93** | 39.00 | 39.50 | 21.50 |
| | ODIN | 28.94 | 38.20 | 32.86 | 56.49 | 20.76 | **22.75** | 25.05 | 38.38 | 36.57 | 41.01 | 36.25 | 22.74 | 40.05 | 31.0 | **20.45** |
| CIFAR-100 | Ours | 50.03 | 36.87 | **13.10** | 16.80 | 45.75 | 37.45 | 37.07 | 46.87 | **16.06** | 66.27 | 15.32 | **18.41** | 60.50 | 25.50 | **14.0** |
| | ODIN | 64.82 | 8.74 | 26.44 | 20.55 | 5.75 | 73.7 | 55.17 | 15.60 | 29.23 | 71.52 | 7.70 | 20.78 | 72.5 | 6.0 | 21.5 |

Table 5: The comparison of our approach with ODIN on black-box adversarial examples

## A.5  VISUALIZATION OF FEATURE SPACE

As seen in Figure 7, we find that the over-generalization reduction leads to a more expressive feature space where all natural out-distribution samples along with many black-box adversarial examples are separated from in-distribution samples to be classified as belonging to the *dustbin* class. Further, some adversarial instances are even placed very close to their corresponding true class, leading the augmented CNNs to classify them correctly.

## A.6  ADVERSARIAL GENERATION METHODS

Generally, an adversarial generation method can either be targeted or untargeted. In targeted attacks, an adversary aims to generate an adversarial sample that makes a victim CNN misclassify it to an adversary selected target class (*i.e.,* $\arg\max F(x + \delta) = y'$, where $y'$ is the targeted class and $\neq y^*$ the actual class). In an untargeted attack, an adversary aims to make the victim CNN to simply misclassify perturbed image to a class other than the true label (*i.e.,* $\arg\max F(x + \delta) \neq y^*$, where $y^*$ is the true class). Here, we briefly explain some well-known

targeted and untargeted attack algorithms.

**Targeted Fast Gradient Sign (T-FGS) (Kurakin et al., 2016a):** This targeted attack method tends to modify a clean image $x$ so that the loss function is minimized for a given pair of $(x, y')$, where target class $y'$ is different from the input's true label $(y' \neq y^*)$. To this end, it uses the sign of gradient of loss function as follows:

$$x_{adv} = x - \epsilon.sign(\nabla J(F(x, \theta), y')), \tag{1}$$

where $J(F(x, \theta), y')$ is the loss function and $\epsilon$ as the hyper-parameter controls the amount of distortion. The transferability of T-FGS samples increases by utilizing larger $\epsilon$ at the cost of adding more distortions to the image. Moreover, the untargeted variant of this method called **FGS** (Goodfellow et al., 2014) is as follows:

$$x_{adv} = x + \epsilon.sign(\nabla J(F(x, \theta), y^*)). \tag{2}$$

**Iterative Fast Gradient Sign (I-FGS) (Kurakin et al., 2017):** This method actually is an iterative variant of Fast Gradient Sign (also called Projected Gradient Descent (PGD) Madry et al. (2017)), where iteratively small amount of FGs perturbation is added by using a small value for $\epsilon$. To keep the perturbed sample in $\alpha$-neighborhood of $x$, the achieved adversary sample in each iteration should be clipped.

$$
\begin{aligned}
x_{adv}^0 &= x \\
x_{adv}^{k+1} &= clip_{x,\alpha}\{x_{adv}^k + \epsilon.sign(\nabla J(F(x_{adv}^k, \theta), y^*))\},
\end{aligned}
\tag{3}
$$

Compared to FGS, I-FGS generates more optimal distortions.

**DeepFool (Moosavi Dezfooli et al., 2016):** This algorithm is an iterative but fast approach for creation of untargeted attacks with very small amount of perturbations. Indeed, DeepFool generates sub-optimal perturbation for each sample where the perturbation is designed to transfer the clean sample across its nearest decision boundary.

**Carlini Attack (C&W) (Carlini and Wagner, 2017b):** Unlike previous proposed methods which find the adversarial examples over loss functions of CNN, this method defines a different objective function which tends to optimize misclassification as follows:

$$f(x') = \max(\max\{Z(x')_{y'} - Z(x')_{y^*}\}, -\kappa) \tag{4}$$

Here $Z(x)$ is the output of last fully connected (before softmax) layer and $x'$ is perturbed image $x$. Also $\kappa$ denotes confidence parameter. A larger value for $\kappa$ leads the CNN to misclassify the input more confidently, however it also makes finding adversarial examples satisfying the condition (having high misclassification confidence) difficult.

**Hyper-parameters of Attack Algorithms:** Each adversarial generation algorithm has a few hyper-parameters as previously seen. We provide details on the hyper-parameters used in our experimental evaluation in Table 6. To generate targeted Carlini attack (called C&W) (Carlini and Wagner, 2017b), we used the authors' github code. Due to large time complexity of C&W, we considered 100 randomly selected images for each dataset. For each selected image, as was done in previous work (Xu et al., 2018), two targeted adversarial samples are generated, where the target classes are the least likely and second most likely classes according to the predictions provided by the underlying CNN. Thus, in total 200 C&W adversarial examples are generated per dataset. To increase transferability of C&W, we utilized $\kappa = 20$ for MNIST and $\kappa = 10$ for CIFAR-10. For CIFAR-100, we used the same setting used for CIFAR-10 except for C&W, we used higher value for $\kappa (= 20)$. For other attacks (variants of FGS and DeepFool), we utilized 2K correctly classified test samples.

| Dataset | Attacks | Parameters & value |
|---------|---------|--------------------|
| MNIST | FGS / TFGS | $\epsilon = 0.2$ |
| | I-FGS | $\epsilon = 0.02$, $\alpha = 0.2$, # of iterations = 20 |
| | C&W | $\kappa = 20$ |
| CIFAR-10 | FGS / TFGS | $\epsilon = 0.03$ |
| | I-FGS | $\epsilon = 0.003$, $\alpha = 0.03$, # of iterations = 20 |
| | C&W | $\kappa = 10$ |
| CIFAR-100 | FGS / TFGS | $\epsilon = 0.01$, # of iteration=6 |
| | I-FGS | $\epsilon = 0.003$, $\alpha = 0.03$, # of iterations = 20 |
| | C&W | $\kappa = 20$ |

Table 6: Adversarial generation methods' hyper parameters

