# OpenReview forum: "Controlling Over-generalization and its Effect on Adversarial Examples Detection and Generation"
_ICLR.cc/2019/Conference_

### Official Review · AnonReviewer2 · 2018-11-05
**Too many hidden assumptions**

**Rating:** 3
**Confidence:** 3

**Review:**

The paper propose to incorporate an additional class for adversarial and out-distribution samples in CNNs. The paper propose to incorporate natural out-distribution images and interpolated images to the additional class, but the problem of selecting the out-distribution images is itself an important problem. The paper presents a very simple approaches for selecting the out-distribution images that relies on many hidden assumptions on the images source or the base classier, and the interpolation mechanism is also too simple and there is the implicit assumption of low complexity images. There exists more principled approaches for selecting out-distribution images that has not considered here like those based on uncertainty estimation or recently proposed direct out-distribution detectors.
In summary, the quality of the paper is poor and the originality of the work is low. The paper is easily readable.

---

> ### Author Response · Authors · 2018-11-15
> **Asking for more explanations**
>
> We are thankful for the reviewer 2 to provide us with his/her feedback,
>
> The reviewer mentioned: "the interpolation mechanism is also too simple":
> We would like to highlight that despite the simplicity of interpolated samples, there has been demonstrated the effectiveness of using such samples on developing more regularized and generalized neural networks (Zhang et al, 2018) as well as on making them more secure (Zhao et. al. 2018).  Thus, we believe that simplicity does not necessarily lead to ineffectiveness.
>
>
> The reviewer mentioned “many hidden assumptions on the images source or the base classier”:
> As this statement is not clear for us, we would appreciate if the reviewer could elaborate more on it. The only assumption we made is on the fact that the out-distribution samples should be statistically and semantically different than the in-distribution samples. Then among such out-distribution sets, we propose a measurement for identifying the most representative one among those available.
>
> The reviewer stated "There exists more principled approaches for *selecting* out-distribution images that has not considered here like those based on uncertainty estimation or recently proposed direct out-distribution detectors":
> While there are the approaches that aim to detect out-distribution sets, they have not been designed for the selection purposes as we do. By mentioning this statement, if the reviewer means the missing of some principled approaches like ODIN in our comparisons, we would like to inform the inclusion of ODIN results in the revised version of the paper.
>
> Reference:
> - Zhao, Jake, and Kyunghyun Cho. "Retrieval-Augmented Convolutional Neural Networks for Improved Robustness against Adversarial Examples." arXiv preprint arXiv:1802.09502 (2018).
>
> - Zhang, H., Cisse, M., Dauphin, Y. N., & Lopez-Paz, D. (2017). mixup: Beyond empirical risk minimization. ICLR 2018 (arXiv preprint arXiv:1710.09412).

---

### Official Review · AnonReviewer1 · 2018-11-06
**The work can be improved**

**Rating:** 4
**Confidence:** 5

**Review:**

The idea of having a separate class for out-distribution is a very interesting idea but unfortunately previously explored. In fact, in machine learning and NLP there is the OOV class which sometimes people in computer vision also use. Some of the claims in the paper can be further substantiated or explored. For example in abstract there is a simple claim that is presented too strong: We also demonstrate that training such an augmented CNN with representative out-distribution natural datasets and some interpolated samples allows it to better handle a wide range of unseen out-distribution samples and black-box adversarial examples without training it on any adversaries. This claim is bigger than just CNNs and needs to be studied in a theoretical framework not an empirical one. Also, one simple way to stop these adversarial cases would be to explore using Sigmoid as opposed to softmax. In general it is very unlikely that you will be able to choose every variation of out-distribution cases. Much easier if you just try to solve the problem using a set of n Sigmoids (n total number of classes) and consider each output a probability distribution.

However, the studies in this paper are still valuable and I strongly recommend continuing on the same direction.

---

> ### Author Response · Authors · 2018-11-15
> **More clarification**
>
> We appreciate the reviewer 1 for his/her feedback on our paper.
>
> The reviewer mentioned: “In general it is very unlikely that you will be able to choose every variation of out-distribution cases”:
> Actually, for training A-CNN (Augmented CNN), we did not train it on every variation of out-distribution cases, rather, we recognize a single representative out-distribution set among the available ones according to our measurement. Then using it for training A-CNN with the aim of effectively controlling over-generalization.
>
>
> The reviewer mentioned: “ Also, one simple way to stop these adversarial cases would be to explore using Sigmoid as opposed to softmax”:
> We would be appreciated if the reviewer could provide us with the references that showing using only sigmoid could control such a challenging problem of adversaries.
> Please note we did not aim to devise a method that is able to reject all adversaries. Rather, we attempted to show that a CNN with less over-generalization is able to reject some of the adversaries while correctly classifies many of the remainder, particularly non-transferable attacks.

---

### Official Review · AnonReviewer4 · 2018-11-14
**Interesting research direction but not good enough**

**Rating:** 4
**Confidence:** 4

**Review:**

This paper proposed to add an additional label for detecting OOD samples and adversarial examples in CNN models. This research direction seems interesting, however, the idea of using an extra label for OODs is not new and was previously explored in different domains. I would expect the describe how their method is different, and keep the research from that point.
Additionally, there are several claims in this paper which I'm not convinced are true, such as the over-generalization of CNNs, the choice of OODs (recent studies have shown NNs are not well calibrated, so using softmax as the confidence might not be the best idea), etc.
Reg. the results, did the authors compare their method to existing adv. example detection methods, such as Ma, Xingjun, et al. ICLR (2018) "Characterizing adversarial subspaces using local intrinsic dimensionality." ? or some other method?
Moreover, in Table 2. I'm not sure what should I conclude from the "Naive Model Error" on OOD samples.

---

### Public Comment · (anonymous) · 2018-10-11
**Confusing the over-generalized regions**

Hi, thanks for the nice paper.

This paper makes a claim:
>>such that  the samples drawn from many over-generalized regions including a wide-range of out-distribution samples and various types of adversaries are mapped to this “dustbin” sub-manifold.

Your theory seems to assume that adversarial examples exist in the over-generalized regions but not in the in-distribution region, am I right? What is the relationship between the over-generalized regions and the in-distribution regions?

---

> ### Author Response · Authors · 2018-10-13
> **To respond the anonymous commenter**
>
> We greatly thank the anonymous commenter,
> Through statistical testing, Grosse et al. [1] demonstrated that adversarial examples are indeed drawn from a distribution different from the distribution of original (in-distribution) data of a given task. Moreover, the over-generalized (i.e. out-distribution) regions contain the samples statistically/semantically different from in-distribution ones. Thus, adversarial examples can be treated as out-distribution samples. We show that if the over-generalized regions can be effectively reduced, then the risk of being fooled by adversaries can be significantly reduced.
>
> In other words, by training an augmented CNN with the dustbin class containing only representative natural out-distribution samples, we tend to refine the frontiers (boundaries) of our model regarding out-distribution regions, and as such improving how adversarial samples are processed, by either correctly classifying these samples or classifying them as dustbin (equivalent to rejection).
>
> Reference: Grosse, Kathrin, Praveen Manoharan, Nicolas Papernot, Michael Backes, and Patrick McDaniel. "On the (statistical) detection of adversarial examples." arXiv preprint arXiv:1702.06280 (2017)

---

> > ### Public Comment · (anonymous) · 2018-10-13
> > **Adversarial examples are not drawn from a different distribution**
> >
> > It should be noted that the work of Grosse et al. was refuted by https://arxiv.org/abs/1705.07263 which shows that it is only adversarial examples generated with fixed and very simple attack algorithms are statistically detectable as different. However, any recent optimization-based attack (PGD/C&W) will generate adversarial examples that are not statistically different from clean examples as far as the detection Grosse et al. scheme will find.

---

> > > ### Author Response · Authors · 2018-10-16
> > > **Our response**
> > >
> > > The incapacity of Grosse et al. approach to detect some well-crafted white-box attacks does not warrant that many of the black-box adversaries, which typically includes more perturbations than their white-box counterparts, are not statistically different from the problem distribution (i.e. in-distribution). For example, for creating transferable black-box C&W adversaries, the amount of perturbations required increases (Carlini and Wagner, 2017).
> > >
> > > Moreover, note that we also exhibit in Table 1 that training an augmented CNN on *only I-FGS* (similar to the approach of Grosse et al.) cannot make it robust to other types of (black-box) adversaries like C&W and DeepFool. Plus, this augmented CNN (trained on I-FGS) is also unable to identify out-distribution samples (Table 3 in Appendix). This may be happened as I-FGS only cover partially the over-generalized regions (Fig. 1 (b)).
> > >
> > > Finally, we are not claiming that our augmented CNN is an ultimate solution for all possible kinds of adversaries. But, through extensive experiments, we shed some light on how controlling over-generalization effectively can positively influence the development of CNNs that are more robust in the presence of *both* natural out-distribution samples and some strong and highly transferable (black-box) attacks.
> > >
> > > Reference :
> > > 1) Nicholas Carlini and David Wagner. Towards evaluating the robustness of neural networks. IEEE Symposium on Security and Privacy (SP), 2017.
> > > 2) Grosse, Kathrin, Praveen Manoharan, Nicolas Papernot, Michael Backes, and Patrick McDaniel. "On the (statistical) detection of adversarial examples." arXiv preprint arXiv:1702.06280 (2017)

---

### Public Comment · (anonymous) · 2018-10-15
**Missing baselines**

You are missing too many baselines.

 I could not see why  out-distribution detection methods can be considered as baselines in the paper.

out-distribution detection:

- baseline: [1]  (Hendrycks & Gimpel, 2016)
- ODIN: [2] (Liang et al., 2017)
- Bayesian Neural Network: [3] (Gal, 2016)
- LearningConfidence: [4] (DeVries & Taylor, 2018)

Similarly,  I could not see why adversarial examples detection methods can be considered as baselines in the paper.

 adversarial examples detection:

- KD+PU: [5]
- LID: [6]


[1] A baseline for detecting misclassified and out-of-distribution examples in neural networks.
[2] Enhancing the reliability of out-of-distribution image detection in neural networks
[3] Uncertainty in deep learning
[4] Learning confidence for out-of-distribution detection in neural networks
[5] Detecting adversarial samples from artifacts.
[6] Characterizing adversarial subspaces using local intrinsic dimensionality.

---

### Public Comment · (anonymous) · 2018-10-17
**Details on Section 2.2**

Hi, could you describe the details on how to find the nearest neighbor of x_i in the feature space of a CNN on Section 2.2 ? Is it expensive computationally for large-scale datasets ?

And is the last layer referring to the previous layer of the logit layer?

---

> ### Author Response · Authors · 2018-11-09
> **Computational complexity**
>
> As mentioned in the paper, the instances are mapped into the feature space generated by the last convolutional layer. We use this space to identify the nearest neighbor using an Euclidean distance, i.e., $min_x’ \|\phi(x) - \phi(x’)\|_2$, where $\phi(x)$ is the representation of $x$ in the feature space. As the dimensionality in the feature space is significantly lower than the original pixel space, the computational complexity of finding nearest neighbors in the feature space is lower than that of the pixel space. By using a proper data structure (e.g., k-d tree, which is implemented in scikit-learn), the complexity time of finding the nearest neighbor can be further reduced, compared to that of a naive k-NN implementation. Moreover, using GPUs and parallel computations, it has been shown the computational time for k-NN can be diminished significantly (Johnson et. al., 2017).
>
> Reference: Johnson, Jeff, Matthijs Douze, and Hervé Jégou. "Billion-scale similarity search with GPUs." arXiv preprint arXiv:1702.08734 (2017)

---

### Public Comment · (anonymous) · 2018-10-22
**Some questions**

>> the fooling classification regions (spanned by the adversary direction and one of its orthogonal random directions) of the naive CNNs are occupied by dustbin regions  (in Section 3.1)

It doesn't mean that  the fooling classification regions of the Augmented CNNs are occupied by dustbin regions. Could you plot several church-windows where the x-axis of each window is the adversary direction achieved by FGS or DeepFool using  the Augmented CNNs?

For Section 3.2, have you tried the targeted adversarial attacks to skip over some regions assigned to dustbin class ?

---

> ### Author Response · Authors · 2018-11-09
> **dustbin regions in white-box adversaries directions**
>
> Question 1 (fooling regions vs dustbin): From Figure 6, it can be observed that moving in the white-box adversaries directions can end up in the dustbin regions. This can show that some of the fooling regions are assigned to the dustbin label in augmented CNNs.
>
> Question 2 (about Sec. 3.2): To generate T-FGS adversaries, we discard the dustbin class as possible fooling target. Adversaries are then generated with a specific epsilon and within a fixed number of iterations (same values as for generating black-box adversaries, see Table 5 of Appendix). To skip over the dustbin regions, the number of iterations for finding adversaries should be increased, which in turn will increase the amount of perturbation.

---

### Public Comment · (anonymous) · 2018-10-26
**Rethink the Interpolated Instances**

The interpolated  samples $ {x}' = \alpha x_i + (1 − \alpha ) x_j $  are not necessarily around the decision boundaries.

Because the interpolation operation is performed in the input space, but not in the high-level feature space.  Assume that the last convolution layer is f(x), $ f({x}') $ is not equal $ f(\alpha x_i + (1 − \alpha ) x_j) $ in fact.

---

> ### Author Response · Authors · 2018-11-09
> **Response**
>
> As you mentioned, we agree that the interpolated samples are not necessarily located on (around) the decision boundaries. There are some reliable approaches for providing an exact solution for this problem, such as DeepFool, that are able to find the samples (adversaries) located around the decision boundaries, but these are computationally expensive.
>
> Zhang et. al (2017) also have considered the interpolated samples (in input space) generated with \alpha=0.5 from the pairs of samples (x_i, x_j) selected from different classes are placed around some virtual decision boundaries as their labels are regarded as the average of their true labels (one-hot vectors i.e. 0.5 y_i + 0.5 y_j ). By regarding such interpolated samples on some virtual decision boundaries, we labeled them as dustbin class.
>
> Zhang, H., Cisse, M., Dauphin, Y. N., & Lopez-Paz, D. (2017). mixup: Beyond empirical risk minimization. ICLR 2018 (arXiv preprint arXiv:1710.09412).

---

### Author Response · Authors · 2018-11-15
**On the originality/novelty of the paper**

The originality of our paper is not on making use of a dustbin class (OOV). As mentioned by the reviewers, this has been proposed not only for NLP and vision tasks but also for adversaries detection (Gross et. al, 2017, Hosseini et. al, 2017) and out-distribution identification (Yu et. al. 2017).

Our main goal is rather to demonstrate the relationship between over-generalization induced by naive CNN and its sensitivity to **both** adversarial examples and out-distribution samples. To this end, we used Augmented CNN (A-CNN) as a tool for controlling over-generalization. Indeed, to our knowledge, we are the first to demonstrate this relationship, i.e. over-generalization and sensitivity to adversaries (as well as out-distribution samples), in an extensive experimental setup. Thus, the originality of our paper does not lie on re-proposing A-CNNs, but we used it as a tool to show that mitigating the effect of over-generalization on the development of more secure neural networks (e.g. CNN) in the presence of adversaries and out-distribution sets.

The immediate natural key question is how to acquire the training samples for the extra class (from a nearly infinite number of out-distribution samples) in order to cover properly the over-generalized regions induced by a naive CNN. Instead of synthesizing artificial out-distribution samples using a hard-to-train generator (Lee. et al ICLR 2018, Jin et. al NIPS 2017, Yu et. al IJCAI 2017 ), we rather propose the use of a novel measurement for selecting a representative out-distribution dataset among the readily accessible natural ones for training an effective A-CNN. This not only maintains the accuracy on in-distribution samples but also results in identifying both adversaries and unseen out-distribution sets simultaneously.

Reference:
- Grosse, Kathrin, et al. "On the (statistical) detection of adversarial examples." arXiv preprint arXiv:1702.06280 (2017).

-Hosseini, Hossein, et al. "Blocking transferability of adversarial examples in black-box learning systems." arXiv preprint arXiv:1703.04318 (2017).

-Jin, Long, Justin Lazarow, and Zhuowen Tu. "Introspective classification with convolutional nets." Advances in Neural Information Processing Systems. 2017.

-Kimin Lee, Honglak Lee, Kibok Lee, and Jinwoo Shin. Training confidence-calibrated classifiers for detectingout-of-distribution samples. ICLR2018.

- Yu, Yang, et al. "Open-category classification by adversarial sample generation." Proceeding of the 26-th International Joint Conference on Artificial Intelligence (2017).

---

### Meta-Review · Area_Chair1 · 2018-12-17
**reject**

**Confidence:** 5
**Recommendation:** Reject

**Metareview:**

The reviewers agree the paper is not ready for publication at ICLR.